# Illness Experiences and Attitudes towards Medication in Online Communities for People with Fibromyalgia

**DOI:** 10.3390/ijerph17228683

**Published:** 2020-11-23

**Authors:** Sabrina Cipolletta, Silvia Caterina Maria Tomaino, Eliana Lo Magno, Elena Faccio

**Affiliations:** 1Department of General Psychology, University of Padua, 35131 Padua, Italy; silviacaterinamaria.tomaino@phd.unipd.it (S.C.M.T.); eliana.lomagno@hotmail.it (E.L.M.); 2Department of Philosophy, Sociology, Pedagogy and Applied Psychology, University of Padua, 35131 Padua, Italy; elena.faccio@unipd.it

**Keywords:** e-health, fibromyalgia, illness experience, medication, online communities

## Abstract

Fibromyalgia is a chronic disabling syndrome, and the legitimacy of its diagnosis is still debated. Internet and online communities may become a relevant resource for affected people. This present study aims to understand the role of online communities relating to fibromyalgia syndrome (FMS) patients’ illness experiences and their attitudes towards medication. A qualitative content analysis based on the grounded theory approach was conducted on 19 conversations from an online forum, and 14 online interviews. Illness experience, lack of reference points, online communities, personal role and attitude towards medication were the five categories identified, with the search for recognition as the core category. The study highlighted that online communities represent a resource that allows users to express and share their needs, especially in terms of legitimacy and recognition.

## 1. Introduction

Fibromyalgia syndrome (FMS) is characterized by chronic widespread musculoskeletal pain, stiffness, sleep disturbance and psychological distress [1]. Even though the syndrome was identified much earlier, diagnostic criteria were not developed until the 1990s [2]. FMS has a global mean prevalence of 2.10%, a European mean prevalence of 2.40% [3] and is more common among females (odds ratio 3:1—F:M) with a variation between different countries [1,4]. The debate regarding the diagnosis’s legitimacy is still open, and widespread skepticism persists. FMS, together with chronic fatigue syndrome (CFS), repetitive strain injury, and irritable bowel syndrome, is a “contested” diagnosis [5], with resulting difficulties for patients, especially in terms of interacting with health professionals and symptom management.

People with FMS generally report a high prevalence of demographic and psychological factors such as a low level of education, failure to complete high school, divorce status [6], high level of somatization [7,8], sleep disturbance and many stressful life events [9,10], a story of emotional trauma [11,12,13], and psychosocial vulnerability [14]. FMS is associated with several mood spectrum disorders, therefore a link with the neuronal functioning system was hypothesized [15], probably involving pain perception.

Chronic illnesses have been investigated since the 1960s, when researchers started to explore social and demographic variables such as coping, stress and the quality of life experienced by people affected by polio, cancer, visible disabilities and terminal experience [16]. From that moment on, the studies’ focuses have extended to illness experience, personal image reorganization, and the daily life impact of the disease experienced by the people affected [17,18]. 

Nowadays, the Internet represents the main source of medical information available. Almost 97% of Internet users refer to “doctor Google” to collect information about health, and are easily directed to forums dealing with reference pathologies [19]. A key component of eHealth is patients’ online communities. Previous studies have shown that forums and online communities can be helpful to share information about personal experiences, medication, treatments and seeking to exchange information with others for people dealing with chronic illnesses [20,21]. Forum users seek support and empathy, which has shown to benefit the posters more than the lurkers [22]. That said, the exchange of information in forums and online communities represents a chance for users to integrate information from many sources [23], or else it represents the risk of assuming the role of a “patient-consumer” that consults online forums or “doctor Google” taking the risk of self-medication and online diagnosis, resulting in potential dangers and limited access to professional care [19]. Another important cause of distress for people with chronic conditions is the excessive health-related internet searching to retrieve information about symptoms, medical conditions and illnesses, things that could easily increase health anxiety, catastrophizing of pain and symptoms, which determine an increased psychological distress and cyberchondria [24]. Online groups, as mixed-sources information containers, could cause psychological distress in participants by increasing health anxiety and the tendency to catastrophize, thus negatively influencing users’ interactions with e-Health resources and exposing them to the danger of misinformation, self-diagnosis and reduced trust in health professionals. Online communities are particularly useful when they are led by a health professional moderator, or when they are included in a complex health information program, rather than communities that provide only peer-to-peer help [25]. Although some studies have shown that offline communities are more effective than online ones in terms of support [26], other studies have shown that online communities make it possible to express the need for help more than is the case with offline communities [21].

Thanks to previous studies [5,19,27] we know more about the use of online communities by people with FMS, especially in terms of the reason for use, topics mainly asked about and participation frequency. Of special interest is Barker’s work [19] that indicated the use of electronic support groups by people with FMS needing to see the syndrome as medically recognized. The present study aims to understand the role of online communities for people with FMS in terms of patients’ illness experiences and their attitudes towards medication, exploring to what extent online peer communities can respond to their needs of legitimation and emotional containment. This is the first study to explore this relationship through a thematic analysis that combines data from online communities and interviews. Expanding knowledge in this field could foster the development or implementation of e-Health digital tools for patients with FMS, resulting in FMS and chronic patients’ needs being more appropriately addressed when facing online tools to manage their syndrome.

## 2. Materials and Methods

### 2.1. Data Collection

This study was based on a content analysis of 21 conversations (616 answers, 75,116 words) among 76 users with FMS that were posted on an Italian forum from December 2017 to October 2018, and on 14 online interviews with users of a Facebook group for people with FMS, created ad hoc by the researchers.

The forum was selected among the few retrieved forums in Italian for patients with FMS. The community is connected to a nationwide patients’ association with a high number of members (4661 members subscribed as of April 2018). The conversations chosen were among the ones with a higher number of views (from 50,604 to 1283) and answers (from 56 to 14) specifically dealing with the FMS experience. The only demographic data we could derive from the analyzed conversations were: gender (34 females and 14 males), average age (34.29; standard deviation was 9.13 years), average years of illness (7.095, ranging from 1 week to 42 years), an average interventions per person (6.53, ranging from 1 to 86), and that 19 participants started a conversation.

Registration to the Facebook group required users to create a nickname to protect privacy and personal data. Within this group, 14 participants were recruited for the online interviews, with the content use being authorized via written informed consent. The 14 online interviews conducted via chat were analyzed for a total of 12,819 words; gender, 13 females and 1 male; average age of 45.5 years (ranging from 29 to 55 years); and average years of illness, 8 years (ranging from 2 to 22 years).

After selecting the forum, we requested a formal authorization from the coordinator, who was responsible for the data protection, to analyze the posts. Both the authors of posts and the interviewees remained anonymous, and were informed that participation was voluntary, revocable at any time, and that privacy was assured. The Ethics Committee of the School of Psychology at the University of Padova (Italy) approved the study (protocol N. 2750).

### 2.2. Data Analysis

A qualitative analysis of the forum conversations and online interviews was completed using the grounded theory approach [28]. The main idea of this approach is that theories should be developed from (i.e., grounded in) the empirical data and subsequent analysis. During data collection, the three steps for realizing a grounded theory were completed. In the first step (open coding), a series of labels (codes) were created by identifying key points from the text. In the second step (axial coding), codes were grouped together in wider categories (macro-categories) and in the third step (selective coding), links between categories and macro-categories were found to identify a core category. The core category represents the research’s main theme and may be defined by an already existing category or by a more abstract term. The results from analyzing the forum and the interviews were put together, resulting in a unitary core category. The software Atlas.ti8 (ATLAS.ti Scientific Software Development GmbH) was used for analysis.

## 3. Results

Five macro-categories were identified: illness experience, lack of reference points, role of online communities, personal role and attitude towards medication. The core category identified was search for recognition. To give a general context, each reported quotation from the online interviews is followed by the identification number of the interview, gender and age of the participant written in parenthesis. Simultaneously, the forum’s quotations are followed by an abbreviation of the respondent’s nickname in parenthesis.

### 3.1. Illness Experience

This category consists of participants’ personal elaboration of their illness, which gave us an overview of the strategies adopted to integrate FMS into their lives. Five subcategories were identified: impact of the illness on the patient’s life, illness history, personal explanations, others’ opinions and relationship with physicians.

#### 3.1.1. Illness’ Impact on the Patient’s Life

Participants mainly expressed the illness’ impact by focusing on the symptoms and the pain felt, e.g., *“If I have to define the last year with a word I would use ‘pain’”* (Ar). In the forum, the participants share different patterns of symptoms. Some report that the suffering is continuous: “*There is no a longer time when I have no pain, I live with it, I cannot say that I have moments without it*” (I3, F, 50), while others report acute symptoms: “*I stay in bed at least two days without any strength because of the pain*” (I14, F, 29). Symptom are reported as being highly incapacitating: “*I am crumpled with pain, I struggle to work and I can’t manage a normal life*” (I9, F, 50). Those limitations may lead to retreating from social life, which negatively impacts the social support available and perceived quality of life: “*I moved away from many people inevitably because I was struggling to do many ’normal’ things. I broke up with friends unfortunately and I realized how difficult it was for me to live in such an uncomfortable situation*” (I13, F, 29). The general experience reported is a complete negative twisting of the life they used to live. Participants try to manage these changes in different ways: listening to and respecting the needs of their body: “*I take the time to do things, not in a hurry*” (I13, F, 29), or trying to manage the pain to continue their activities at the same pace as always.

#### 3.1.2. Illness History

Most of the participants that reported suffering from FMS for many years also reported an embedded personal illness history previous to FMS onset, characterized by the presence of previous symptoms or diseases -both related and unrelated to FMS- and other traumatic events: “*My mother has been suffering from FIBROMYALGIA for many years (…) I was afraid they would tell me ‘Miss, you have fibromyalgia,’ a great big fear that… has come true.*”(Ta). Participants reported the illness’ duration may vary depending on personal experience: “*I have suffered from muscular pains since I don’t know how many years precisely, maybe 6, but they have worsened in the last 2 or 3 years…*” (Ta). 

In contrast, some participants reported a more recent history of illness related to the onset of FMS: 

“*I have never had any physical problems, only the usual temporary things that everyone has (like occasional headaches). (…) But then last year things changed, and I slowly fell into a nightmare. It all started with pain in the knees, first one and then the other. Permanent, chronic, continuous*” (Nu). 

Some reported that they initially ignored the symptoms, which were perceived to be temporary (e.g., “*the change of season,*” I3, F, 50). Meanwhile, FMS developed into an increasingly persistent and widespread syndrome: “*I have always had some symptoms but I didn’t pay any attention (…) However, around February, following a rather stressful period (…) it became increasingly difficult to face the day and go to work*” (I1, F, 44).

#### 3.1.3. Personal Explanations

This category contains participants’ attempts to make sense of their illness. The explanations mainly are attributed to medical reasons: “*maybe something in the DNA*” (I7, F, 49) or psychological, mainly personal characteristics or emotional trauma: “*FM is caused by a trauma from being late in a situation and not being able to prevent a sad situation, an accident or a grief, or a problem, whatever it may be*” (E2010). There are differences between participants’ responses when asked to explain FMS in general, and when they are asked to explain their personal experiences of FMS: In the first situation, participants look for medical explanations, whereas in the second situation they very often mention a specific episode in their life they consider as the triggering event (e.g., chronic stress, pregnancy, mourning, etc.), combining together information from different sources such as the Internet, physicians, and word of mouth. 

#### 3.1.4. Others’ Opinions

Users reported that fibromyalgia commonly is not perceived as a real illness, and the pain that patients suffer is considered to be "*imaginary*" (I7, F, 49); “*My parents still don’t believe me (…) they still don’t believe everything*” (I10, M). This skepticism leads others to identify the participants as being "*weak people*" (I2, F, 48). Simultaneously, people tend to underestimate the symptoms: "*Strangers pretend to understand, but those who do not live with a person with fibromyalgia cannot understand!*" (I7, F, 49). In a few cases, the participants’ relatives changed their minds: “*They all thought I was a hypochondriac, that I was exaggerating. Slowly, the people who really love me (…) realized that I wasn’t inventing anything*” (Ta). In other cases, relatives’ opinions did not change, and indeed negatively influenced their behavior: “*Others either do not believe that it is possible to suffer so much… or even after a while they get tired of your outbursts, put aside understanding, and decide that it is not their problem.*” (Ar).

#### 3.1.5. Relationship with Physicians

Physicians represent the reference person for patients, and their behavior is detected using two codes: centrality of the patient during the visit and prescriptions. Participants often reported experiences related to medical visits by highlighting their doubts and fears: “*The doctor did not take any account of the symptoms I reported and looked at the tests only superficially (…). In the end she told me to ‘look for a psychologist because your symptoms are psychosomatic’*” (Gi73). Some participants reported feeling confident and satisfied with doctors who paid special attention in the care process: “*The doctor told me he didn’t want to give me drugs for now because I’m young and I’m in a more or less initial phase of fibromyalgia. He told me to try to manage my emotions and to understand my body*” (Ta). Others reported feeling distrustful: “*For two years a presumptuous primary care physician who thought he knew more than all the specialists made me lose precious time*” (F, 50).

### 3.2. Lack of Reference Points

Many of the participants’ experiences referred to lacking reference points, which can lead to uncertainty and a variety of emotions such as confusion, anger, and sadness.

#### 3.2.1. Uncertainty

Uncertainty is a common experience, in fact the diagnostic process is described as a "*calvary*" (I7, F, 48) and is characterized by many contacts with different specialists: "*I started a calvary between orthopedic … neurologists… neurosurgeons and rheumatologists (…) I call it a calvary because it is absurd to be told all the time—I can do nothing. You must go to X—and you go to X who tells you the same thing*” (I9, F, 50). The codes associated with this subcategory are: complex diagnoses, switching from one doctor to another, lack of answers and waiting times. The uncertainty concerns the patient’s detection of their condition and also their interpretation of symptoms, both involving a series of examinations that patients define as “*an odyssey*” (Ar). Patients state that fibromyalgia is diagnosed “*for exclusion*,” by disconfirming other diseases. The need for a permanent diagnosis is central in the participants’ illness experience, because it legitimizes their condition and allows them to receive specific treatments: “*I visited the rheumatology department and the diagnosis was fibromyalgia. In truth, finding out was a relief: the thousand pains had a name.*” Diagnostic processes are complex and contested. Moreover, frequently physicians are not in agreement with each other: “*My family doctor told me that, according to him, I have fibromyalgia (never visited, only based on what I reported and was not convinced of the rheumatologist’s diagnosis)*” (BG). 

#### 3.2.2. Emotions Related to the Lack of Reference Points

Participants feel disoriented about their condition, reporting a lack of reference points and mainly expressing negative emotions such as anger, loneliness, resignation and despair. Anger is mostly addressed in the context of doctors who do not provide the expected professional attitude and who are not able to give answers: “*Obviously at first I experienced anger and bitterness*” (Ma). Despair is related to feeling helpless with regard to the situation and to the feeling of not being properly listened to, which leads to a consequent resignation because of the lack of specific treatment and of proper recognition from the health system: “*I am desperate, what future do I have?*” (BG).

### 3.3. Role of Online Communities 

We identified three codes within the theme of the role of online communities: motivation, user’s role and expectations.

#### 3.3.1. Motivation

The reasons behind registering to online communities are: the search for information, the need to ask for advice from others and the need for recognition of the personal illness experience. Participants complain about rarely finding valid information in the offline world. They use the Internet autonomously to assuage doubts about their symptoms, to get in touch with other members of the community and to get to know more about the efficacy or side effects of the prescribed medications: “*I decided to ask you because I admit to having minimal trust in doctors and would like a comparison with direct experiences*” (Ar). One of the most important reasons for participating in the forum is the possibility of sharing personal experiences with others who can really understand: “*I joined the forum because I often feel like an extra-terrestrial when I talk about my difficulties with someone (…) Here (in the community) I think I can find people who understand how I feel*” (M76).

#### 3.3.2. User’s Role

In terms of the user’s role, participants are mainly readers of posts: “*I am not very active, at the moment I am a spectator, sometimes I answer or write about my experience*” (I10, M). Most of them reported looking for information and other’s personal experiences. Those who participate more actively by posting in the group, mainly ask for advice or post materials they retrieved from different sources to start an exchange of opinions and information: “*I post it here since for sure you have more information than me, I hope that some of you by reading my symptoms could help me, or at least that someone could tell me how to proceed or if the direction I am following is possibly the right one*” (Gi73).

#### 3.3.3. Expectations

Participants believe that the community is a place not just to share personal experiences and complaints, but also to support and provide emotional containment: “*In the meantime, thank you for your interest … makes me feel less alone!*” (Gi73) and “*Yes, in the end it seems to me that therapies are similar, times change but we have tried the same things...Who knows that with so many experiences we cannot find the most effective therapies*” (M76).

### 3.4. Personal Role

Exploring self-narratives, illness is reported as an invalidating experience that affects mostly personal independence in everyday life. In general, patients affected by FMS tend to describe themselves by comparing their actual situation to the situation lived before the syndrome’s onset. Self-descriptions and offline resources are important components of their personal role.

#### 3.4.1. Self-Description

Participants described themselves mostly by referring to the limitations that FMS has involved or reporting personal characterizations that do not take into account the pathological condition experienced, or including FMS experience in their self-description. These different choices result in three self-description styles, which are characterized by contempt and pessimism, brooding hostility and acceptance.

The self-description characterized by contempt and pessimism involves viewing the disease negatively, such as experiencing it as a catastrophe or punishment. Participants report feeling powerless against it, so much so that the illness’ effects have destroyed their life irreparably: “*There was a before with a certain person who did certain things, and there is an after with a completely different person who is sick and would like to be heard and understood*” and “*I loved dancing but I cannot do it anymore (…) I went to the gym and I cannot do it anymore. You can understand why I am a sad person and cannot be otherwise*” (I7, F, 49). These participants generally report a poor social network, few significant relationships with family, friends and colleagues, together with a perceived sense of inability from their loved ones when it comes to taking care of the physical and psychological suffering related to their FMS.

Brooding hostility derives from needing to affirm the personal role that the FMS condition threatens: “*I am a super mom, a super woman, a very sensitive and caring person for those around me*” (I6, F). These participants describe themselves and their relationship with FMS in terms of the anger they feel for their condition, underlining that they had not surrendered.

Acceptance suggests an optimistic and active response to the illness condition, especially in terms of adaptation and internalization: "*I think I am strong, unfortunately (…) my medical record is wide and vast; at the same time, however, certainly I am emotional and sensitive*” (I3, F, 50). FMS is experienced as a challenge and they try their best to adapt their life’s routine, based on the changes the illness causes: “*FM has upset my lifestyle, my way of approaching life… and for this I am very happy; I believe that nothing comes for nothing and everything has a meaning*” (I13, F, 29).

#### 3.4.2. Offline Resources

Offline resources include the codes social relationships, working life and economic situation. Family mostly represents the main offline help source, in contrast with the lack of comprehension typically experienced in the workplace: “*My family well accepted my condition, and everyone helps me, but where I work I was removed from what I liked doing without any explanation because unfortunately I had to stay at home a few months*” (I6, F). In many cases the work environment can be hostile in not providing understanding or career protection: “*I had to make a choice [leaving work], also because the company doctor did not recognize me a right to reduce working hours*” (I4, F, 49).

### 3.5. Attitude towards Medication

Analyzing interviews and the forum, we highlighted three main attitudes towards medication depending on the type of intake and the reason for intake: rejection, preferential intake and only in need intake.

#### 3.5.1. Rejection

Some participants tend to refuse any type of medication (pills, manual therapy, etc.), considering them ineffective. These people feel hopeless and tend to appear passive regarding managing their illness. Generally, they refer to distrusting physicians and experiencing FMS as a “punishment.”

#### 3.5.2. Preferential Intake 

Most participants indicate that taking medication is the best and fastest solution to facing pain and other symptoms. These people usually manage their medications independently, which leads to frequently changing the type of medication used to find the most efficient one when it comes to reducing pain and the illness’ limitations to regularly conduct their lives (going to work, being present in the family, etc.).

#### 3.5.3. Only in Need Intake

Some participants take medication only when needed, specifically when pain becomes unbearable. Generally, these people consider different types of medication, and some of them report that they prefer taking comparative and alternative medicine, rather than traditional drugs.

### 3.6. Search for Recognition

The search for recognition is the core category containing all the others that emerged from the grounded theory analysis. To date, FMS is not fully legitimated because of poor etiopathogenic knowledge and the poor medical consideration given to this syndrome [7]. The misrecognition of the syndrome and its impact turns into a lack of a right to health for the people affected by FMS, such as: lack of support and facilitation measures, appropriate drug prescription and level of assistance exemption. Online communities allow people affected by FMS to feel recognized by other people living the same experience, which can facilitate an active request for recognition, for the legitimacy of the syndrome itself, for health rights and for appropriate medical assistance [21].

Figure 1 shows the links between the core category and the other categories: starting from a subordinate level, the attitude towards medication depends on the personal role and is associated with the illness experience. Specifically, a personal role centered on contempt leads to rejecting any medication, brooding hostility towards preferential medication use, and acceptance towards medication only when needed. A lack of reference points contributes to the illness experience and leads to the participation in online communities. Personal role, illness experience and online communities are all associated and can be connected to the search for recognition.

## 4. Discussion

The present study’s results showed that patients’ experiences of FMS are connected to different personal roles resulting in peculiar uses of the online community and resources, which opens possible horizons for further developing tailored Internet-based intervention and online resources for patients with FMS [5,21]. In fact, participants accepting their personal FMS condition tend to address other users and online resources in a proactive way, not only as a platform for finding information and clinicians to consult, but also as a resource to connect and share personal experiences, the sense of togetherness with other patients and to explore other possible therapies or resources [5]. However, participants reporting an FMS experience characterized by pessimism and hostility feel deprived of their personal role because of the illness, thus they tend to address the online community in a passive way, such as by looking to find compassion and sympathy from other users or by expressing their anger and negative feelings. This attitude results in behaviors such as lamenting, not exploring other possibilities and relating to online communities as tools to ask for information. These last results represent a new type of data from the present study.

This study has highlighted that patients affected by FMS interacting within online communities report the need for control over the syndrome and treatment. In line with previous studies [29], the participants in this study expressed the need for receiving a permanent diagnosis as the first step to addressing the illness, and searching for help and requiring a more accurate medical and legal recognition of the syndrome. The patients expressed a strong willingness to learn more about their condition and the Internet is often the main source of such information. Online community users claim to have registered to share experiences and advice, especially regarding specialists to consult, diagnostic assessment and medication.

The lack of valid information suggests that designing an online platform for FMS, incorporating medical information, treatment and specialists, administrated by a qualified professional on FMS and online communities is strongly required. According to the literature [5,27,30], online communities represent a valuable resource for emotional and personal support; as reported by our participants, the online peer community is perceived as a safe space where they can share emotions and fears, and look for support and a sense of togetherness. A formal FMS community should develop online and offline intervention program, both for patients and health professionals, to provide support when it comes to facing the illness process—from assessment to everyday treatment.

Tailored digital interventions aiming to improve health care for patients with a chronic condition are strongly required in the modern healthcare scenario; in fact, as supported by the literature [31,32], e-Health interventions, Internet-based interventions and Internet-based peer support programs may present numerous benefits in treating chronic conditions. Accordingly, cost-effectiveness, flexible accessibility and having no wait time when using e-Health resources could help patients with chronic conditions to properly access medical and psychological consultation and resources when needed; thus fostering a better engagement and adherence in the care process, adequate access to care, optimized exposure to the intervention and decreased sense of abandonment and incomprehension experienced [33].

Another possible interpretation of our results can be addressed in the context of the theoretical framework of job demand control support model [34]. This model could explain the importance of social support and its quality in online communities in terms of the possible positive or negative influences over personal well-being and perceived support. Social support has a key role in influencing well-being, as suggested by many studies [35]. In fact, a supportive environment has a relevant effect especially in terms of improving psychological and physical outcomes. Thus, people with FMS accepting their condition and actively being involved in the online communities could report an increased sense of well-being and belongingness connected to the perception and experience of a supportive online community. On the contrary, people with FMS experiencing hostility and pessimism for their condition—especially when interacting with an online community mainly characterized by negative emotional sharing and lamenting—could face the risk of being negatively influenced in terms of satisfaction, perception of support and decreased sense of well-being. In light of this, fostering a supportive climate in online communities for people with FMS may represent a strategic approach to improve online posters’ and lurkers’ well-being and sense of commitment, thus enhancing the effectiveness of e-Health interventions with FMS patients.

The present study’s main limitation is that the participants interviewed were recruited from an online community created ad hoc a (Facebook group), which is different from the one we extracted conversations from (the Forum), thus resulting in a significant heterogeneity of data considered for analysis. Consequently, it was not possible to compare forum users’ behavior with the knowledge derived from the interviews. Future research might fill this gap by directly interviewing the users of the same community under analysis.

## 5. Conclusions

This study aimed to explore the role of online communities for people suffering from FMS. The results showed that online communities are efficient tools for giving voice to the collective need for legitimacy for one’s illness experience and for recognizing the right to health.

Therefore, the community acts as a motor to share precious information about diagnostic processes, different treatments and specialists to consult. It provides peer-to-peer emotional and practical support, validating the personal illness experience. Further investigations are needed to develop a model of a valid and comprehensive online community for patients with FMS, suitable for their personal, medical and practical needs.

## Figures and Tables

**Figure 1 ijerph-17-08683-f001:**
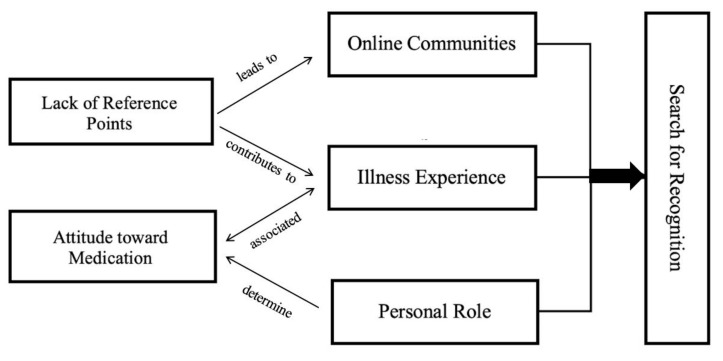
Model derived from the grounded theory analysis.

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
