# Peer review of "Illness Experiences and Attitudes towards Medication in Online Communities for People with Fibromyalgia"

_ijerph, 2020, doi:10.3390/ijerph17228683_

Round 1
Reviewer 1 Report
The theme is interesting, and the manuscript is overall well written and well presented. Some minor concerns have to be solved.
The introduction is of good quality. Theoretical background is clear and well supported by relevant references on the theme. I suggest to state in a more explicit way the literature gap you want to bridge with your study.
The methods section is overall of good quality. All the methods are clearly described and supported by relevant scientific literature.
The discussion is well performed. An interesting part of the results is that patients experiences are connected to different personal roles resulting in peculiar uses of the online community and resources. I suggest to frame these results in the theoretical models such as Job Demand Control Support model. Are your results in line with such theorizations? Moreover, some of those aspects could represent a useful target to address improving strategies, in order to manage help programs for people suffering FMS. For example, improving job support and social relations could lead to mitigate the effect of FMS. Authors can also refer to these updated scientific papers:
Wu G, Wu Y, Li H, Dan C. Job Burnout, Work-Family Conflict and Project Performance for Construction Professionals: The Moderating Role of Organizational Support. Int J Environ Res Public Health. 2018 Dec 14;15(12):2869. doi: 10.3390/ijerph15122869. PMID: 30558218; PMCID: PMC6313469.
Lecca, L.I., Finstad, G.L., Traversini, V., Lulli L.G., Gualco, B., Taddei, G. The role of job support as a target for the management of work-related stress: The state of art. Quality - Access to Success, 2020, 21(174), pp. 152-158
Kowalczuk K, Krajewska-Kułak E and Sobolewski M (2020) Working Excessively and Burnout Among Nurses in the Context of Sick Leaves. Front. Psychol. 11:285. doi: 10.3389/fpsyg.2020.00285
In the conclusion section you can state what this paper adds to the knowledge about the theme.
Best Regards
Author Response
- The introduction is of good quality. Theoretical background is clear and well supported by relevant references on the theme. I suggest to state in a more explicit way the literature gap you want to bridge with your study.
Response: As suggested, authors updated the aim of the study in the last lines of the section 1.Introduction: “The present study aims to understand the role of online communities for people with FMS in terms of patients’ illness experiences and their attitudes towards medication, exploring to what extent online peer communities can respond to their needs of legitimation and emotional containment. This is the first study to explore this relationship through a thematic analysis that combines data from online communities and interviews. Expanding knowledge in this field could foster the development or implementation of e-Health digital tools for patients with FMS, resulting in FMS and chronic patients’ needs being more appropriately addressed when facing online tools to manage their syndrome.”
- The methods section is overall of good quality. All the methods are clearly described and supported by relevant scientific literature.
The discussion is well performed. An interesting part of the results is that patients experiences are connected to different personal roles resulting in peculiar uses of the online community and resources. I suggest to frame these results in the theoretical models such as Job Demand Control Support model. Are your results in line with such theorizations? Moreover, some of those aspects could represent a useful target to address improving strategies, in order to manage help programs for people suffering FMS. For example, improving job support and social relations could lead to mitigate the effect of FMS. Authors can also refer to these updated scientific papers:
Wu G, Wu Y, Li H, Dan C. Job Burnout, Work-Family Conflict and Project Performance for Construction Professionals: The Moderating Role of Organizational Support. Int J Environ Res Public Health. 2018 Dec 14;15(12):2869. doi: 10.3390/ijerph15122869. PMID: 30558218; PMCID: PMC6313469.
Lecca, L.I., Finstad, G.L., Traversini, V., Lulli L.G., Gualco, B., Taddei, G. The role of job support as a target for the management of work-related stress: The state of art. Quality - Access to Success, 2020, 21(174), pp. 152-158
Kowalczuk K, Krajewska-Kułak E and Sobolewski M (2020) Working Excessively and Burnout Among Nurses in the Context of Sick Leaves. Front. Psychol. 11:285. doi: 10.3389/fpsyg.2020.00285
Response: As suggested by the reviewer, the JCDS model was taken into account. Authors do find it suitable to explain the role of social support in online communities as an important resource to improve posters’ and lurkers’ psychological and physical well-being. The subsequent part has been added to the discussion in order to clarify it: “Another possible interpretation of our results can be addressed in the context of the theoretical framework of Job Demand Control Support Model [34]. This model could explain the importance of social support and its quality in online communities in terms of the possible positive or negative influences over personal well-being and perceived support. Social support has a key role in influencing well-being, as suggested by many studies [35]. In fact, a supportive environment has a relevant effect especially in terms of improving psychological and physical outcomes. Thus, people with FMS accepting their condition and actively being involved in the online communities could report an increased sense of well-being and belongingness connected to the perception and experience of a supportive online community. On the contrary, people with FMS experiencing hostility and pessimism for their condition—especially when interacting with an online community mainly characterized by negative emotional sharing and lamenting—could face the risk of being negatively influenced in terms of satisfaction, perception of support and decreased sense of well-being. In light of this, fostering a supportive climate in online communities for people with FMS may represent a strategic approach to improve online posters’ and lurkers’ well-being and sense of commitment, thus enhancing the effectiveness of e-Health interventions with FMS patients.”
[34] Karasek, R., and Theorell, T. (1990). Healthy work: stress, productivity and the reconstruction of working life. New York (N.Y.): Basic books.
[35] Lecca, L., I., Finstad, G., L., Traversini, V., Lulli, L., G., Gualco, B., Taddei, G. (2020). The role of job support as a target for the management of work-related stress: The state of art. Quality - Access to Success. Journal of Management System, 21(174): 152-158.
Reviewer 2 Report
Authors presents an interesting analysis about the electronic communities of fibromyalgia patients. Some aspects must be considered before this article is suitable for publication, beginning by checking grammar:
The manuscript introduction offers enough information about the topic, however I consider that only the good face of eHealth is shown. I miss information about the possible dangers of eHealth: self-medication, online diagnosis, catastrophization... This information would increase the quality of introduction.
Regarding research design, it must be considered that a mix of forums and Facebook were considered, being this a limitation of the study. It would be desirable to focus on specific Forums (76 users) or on Facebook (14 users), thus the mix of both could result in significant heterogeneity. Anyway, this can not be changed and I consider it is enough to mention it at limitations.
Concerning Results, I recognize that I can not understand the meaning of the codes after each sentence. This is why I asked the authors to explain them before their use, as I imagine this could happen to other readers. Are those codes related to participants' identity or to the forum? An explanation about this is required in my opinion.
Despite this, I insist that this article is very original and analyse a phenomena which is live and that could give the health professionals the key of tomorrow's health promotion.
Line 25: re-write the sentence, as it is grammatically incorrect thus you repeat "and"
Line 32: place each reference next to the point referenced, not at the end of the whole sentence.
Line 60 and 78: check grammar
Line 96: please include Enterprise and country. I algo suggest to include more details about the analysis performed.
Line 103: I suggest using "gived" instead of "gives", as study has concluded.
Line 109: please explain the use of acronyms as (Ar) and other as I3, F, 50). Information to understand this must be included previously to its use.
Lines 123-125: please be concise about the importance of having referred previous traumatic events or not by the participants. You just say that "some did not report previous illness" and the opposite too. Quantify the importance of this data according with how many participants referred each condition or clearly say that there was no differences between people that had suffered traumatic events or not.
Line 163: please check title section
Line 208: check grammar
Line 280: I suppose that it is "Grounded" instead of "Grouned"
Line 281: a reference would be desirable to support this affirmation
Lines 284-287: a reference supporting this affirmation is needed
Figure 1: I recommend to use a higher quality picture, checking "Lack of Reference points" square, where a line over the "L" of "Lack" should be removed
Lines 298-309: some references are needed to support these affirmations.
Line 315: check double space after the point
Author Response
- Authors presents an interesting analysis about the electronic communities of fibromyalgia patients. Some aspects must be considered before this article is suitable for publication, beginning by checking grammar:
Response: A close editorial has been performed by Proofreading.com.
- The manuscript introduction offers enough information about the topic, however I consider that only the good face of eHealth is shown. I miss information about the possible dangers of eHealth: self-medication, online diagnosis, catastrophization... This information would increase the quality of introduction.
Response: As suggested possible dangers of e-Health have been added in the Introduction section, as following: “the exchange of information in forums and online communities represents a chance for users to integrate information from many sources [23], or else it represents the risk of assuming the role of a “patient-consumer” that consults online forums or “Doctor Google” taking the risk of self-medication and online diagnosis, resulting in potential dangers and limited access to professional care [19]. Another important cause of distress for people with chronic conditions is the excessive health-related internet searching to retrieve information about symptoms, medical conditions and illnesses, things that could easily increase health anxiety, catastrophizing of pain and symptoms, which determine an increased psychological distress and cyberchondria [24]. Online groups, as mixed-sources information containers, could cause psychological distress in participants by increasing health anxiety and the tendency to catastrophize, thus negatively influencing users’ interactions with e-Health resources and exposing them to the danger of misinformation, self-diagnosis and reduced trust in health professionals.”
[24] Gibler, R., C., Jastrowski Mano, K., E., O’Bryan, E., M., Beadel, J. R., & McLeish, A., C. (2019): The role of pain catastrophizing in cyberchondria among emerging adults, Psychology, Health & Medicine, DOI: 10.1080/13548506.2019.1605087
- Regarding research design, it must be considered that a mix of forums and Facebook were considered, being this a limitation of the study. It would be desirable to focus on specific Forums (76 users) or on Facebook (14 users), thus the mix of both could result in significant heterogeneity. Anyway, this can not be changed and I consider it is enough to mention it at limitations.
Response: The limitations of the study have been updated considering the Reviewer comment: “The present study’s main limitation is that the participants interviewed were recruited from an online community created ad hoc a (Facebook group), which is different from the one we extracted conversations from (the Forum), thus resulting in a significant heterogeneity of data considered for analysis. Consequently, it was not possible to compare forum users’ behavior with the knowledge derived from the interviews.”
- Concerning Results, I recognize that I can not understand the meaning of the codes after each sentence. This is why I asked the authors to explain them before their use, as I imagine this could happen to other readers. Are those codes related to participants' identity or to the forum? An explanation about this is required in my opinion.
Response: The necessary information to understand in text quotations have been added in the section 3. Results: “To give a general context, each reported quotation from the online interviews is followed by the identification number of the interview, gender and age of the participant written in parenthesis. Simultaneously, the forum’s quotations are followed by an abbreviation of the respondent’s nickname in parenthesis.”
Despite this, I insist that this article is very original and analyse a phenomena which is live and that could give the health professionals the key of tomorrow's health promotion.
- Line 25: re-write the sentence, as it is grammatically incorrect thus you repeat "and"
Response: The sentence was re-written in the following way: “FMS has a global mean prevalence of 2.10%, a European mean prevalence of 2.40% [3] and is more common among females (odds ratio 3:1 - F:M) with a variation between different countries [1, 4].”
- Line 32: place each reference next to the point referenced, not at the end of the whole sentence.
Response: References have been moved as suggested: “People with FMS generally report a high prevalence of demographic and psychological factors such as a low level of education, failure to complete high school, divorce status [6], high level of somatisation [7, 8], sleep disturbance and many stressful life events [9, 10], a story of emotional trauma [11, 12, 13], and psychosocial vulnerability [14].”
- Line 60 and 78: check grammar
Response: The sentence in line 60 was re-written in the following way: “The present study aims to understand the role of online communities for people with FMS in terms of patients’ illness experiences and their attitudes towards medication, exploring to what extent online peer communities can respond to their needs of legitimation and emotional containment.”
The sentence in line 78 was re-written in the following way: “The only demographic data we could derive from the analysed conversations were: gender (34 females and 14 males), average age (34.29; standard deviation was 9.13 years), average years of illness (7.095, ranging from 1 week to 42 years), an average interventions per person (6.53, ranging from 1 to 86), and that 19 participants started a conversation”
- Line 96: please include Enterprise and country. I algo suggest to include more details about the analysis performed.
Response: As suggested, the information requested were added: “The software used for the analysis was ATLAS.ti8 (ATLAS.ti Scientific Software Development GmbH).”
More details about the analysis performed were added, thus the section 2.2 Data Analysis has been rewritten in the following way: “A qualitative analysis of the forum conversations and online interviews was completed using the Grounded Theory approach [28]. The main idea of this approach is that theories should be developed from (i.e. grounded in) the empirical data and subsequent analysis. During data collection, the three steps for realizing a Grounded Theory were completed. In the first step (open coding), a series of labels (codes) were created by identifying key points from the text. In the second step (axial coding), codes were grouped together in wider categories (macro-categories) and in the third step (selective coding), links between categories and macro-categories were found to identify a core category. The core category represents the research’s main theme and may be defined by an already existing category or by a more abstract term. The results from analysing the forum and the interviews were put together, resulting in a unitary core category. The software Atlas.ti8 (ATLAS.ti Scientific Software Development GmbH) was used for analysis.”
- Line 103: I suggest using "gived" instead of "gives", as study has concluded.
Response: As suggested, “gives” was replaced with “gave”.
- Line 109: please explain the use of acronyms as (Ar) and other as I3, F, 50). Information to understand this must be included previously to its use.
Response: The necessary information to understand in text quotations have been added in the section 3.Results: “To give a general context, each reported quotation from the online interviews is followed by the identification number of the interview, gender and age of the participant written in parenthesis. Simultaneously, the forum’s quotations are followed by an abbreviation of the respondent’s nickname in parenthesis.”
- Lines 123-125: please be concise about the importance of having referred previous traumatic events or not by the participants. You just say that "some did not report previous illness" and the opposite too. Quantify the importance of this data according with how many participants referred each condition or clearly say that there was no differences between people that had suffered traumatic events or not.
Response: We have re-written the whole paragraph to be concise and to make it clear that we are not interested in the prevalence of each condition but rather in differentiating among between those who have been suffering from FMS for a long time and those who reported a more recent onset or initially ignored the symptoms: Most of the participants that reported suffering from FMS for many years also reported an embedded personal illness history previous to FMS onset, characterized by the presence of previous symptoms or diseases -both related and unrelated to FMS- and other traumatic events: “My mother has been suffering from FIBROMYALGIA for many years […] I was afraid they would tell me ‘Miss, you have fibromyalgia,’ a great big fear that… has come true.”(Ta). Participants reported the illness’ duration may vary depending on personal experience: “I have suffered from muscular pains since I don’t know how many years precisely, maybe 6, but they have worsened in the last 2 or 3 years…”(Ta).
In contrast, some participants reported a more recent history of illness related to the onset of FMS:
“I have never had any physical problems, only the usual temporary things that everyone has (like occasional headaches). […] But then last year things changed, and I slowly fell into a nightmare. It all started with pain in the knees, first one and then the other. Permanent, chronic, continuous” (Nu).
Some reported that they initially ignored the symptoms, which were perceived to be temporary (e.g. “the change of season,” I3, F, 50). Meanwhile, FMS developed into an increasingly persistent and widespread syndrome: “I have always had some symptoms but I didn’t pay any attention […] However, around February, following a rather stressful period […] it became increasingly difficult to face the day and go to work” (I1, F, 44).
- Line 163: please check title section
Response: Title section was corrected.
- Line 208: check grammar
Response: The sentence was re-written in the following way: “They use the Internet autonomously to assuage doubts about their symptoms, to get in touch with other members of the community and to get to know more about the efficacy or side effects of the prescribed medications.”
- 14. Line 280: I suppose that it is "Grounded" instead of "Grouned"
Response: The typing error “Grouned” was corrected in “Grounded”.
- Line 281: a reference would be desirable to support this affirmation
Response: Reference has been added: “To date, FMS is not fully legitimated because of poor etiopathogenic knowledge and the poor medical consideration given to this syndrome [7].”
[7] Berger, A., Dukes, E., Martin, S., Edelsberg, J., & Oster, G. (2007). Characteristics and healthcare costs of patients with fibromyalgia syndrome. International journal of clinical practice 61(9): 1498-1508. DOI: 10.1111/j.1742-1241.2007.01480.x
- Lines 284-287: a reference supporting this affirmation is needed
Response: Reference has been added: “Online communities allow people affected by FMS to feel recognized by other people living the same experience thing that may facilitate an active request for recognition, for the legitimacy of the syndrome itself, for health rights and for appropriate medical assistance [21].”
[21] Cipolletta, S., Votadoro, R., & Faccio, E. (2017). Online support for transgender people: an analysis of forums and social networks. Health & social care in the community 25(5): 1542-1551. DOI: 10.1111/hsc.12448
- Figure 1: I recommend to use a higher quality picture, checking "Lack of Reference points" square, where a line over the "L" of "Lack" should be removed
Response: Figure 1 has been corrected and a higher quality version of it was inserted in the paper.
- Lines 298-309: some references are needed to support these affirmations.
Response: References have been added where there are similar results in previous studies: “The present study’s results showed that patients’ experiences of FMS are connected to different personal roles resulting in peculiar uses of the online community and resources, which opens possible horizons for further developing tailored Internet-based intervention and online resources for patients with FMS [5, 21]. In fact, participants accepting their personal FMS condition tend to address other users and online resources in a proactive way, not only as a platform for finding information and clinicians to consult, but also as a resource to connect and share personal experiences, the sense of togetherness with other patients and to explore other possible therapies or resources [5]. However, participants reporting an FMS experience characterized by pessimism and hostility feel deprived of their personal role because of the illness, thus they tend to address the online community in a passive way, such as by looking to find compassion and sympathy from other users or by expressing their anger and negative feelings. This attitude results in behaviors such as lamenting, not exploring other possibilities and relating to online communities as tools to ask for information.” These last results are a new type of data from the present study.
[5] Chen, A. T. (2012). Exploring online support spaces: using cluster analysis to examine breast cancer, diabetes and fibromyalgia support groups. Patient education and counseling 87(2): 250-257. DOI: 10.1016/j.pec.2011.08.017
[21] Cipolletta, S., Votadoro, R., & Faccio, E. (2017). Online support for transgender people: an analysis of forums and social networks. Health & social care in the community 25(5): 1542-1551. DOI: 10.1111/hsc.12448
- Line 315: check double space after the point
Response: Done.
Round 2
Reviewer 2 Report
Authors have solved the points suggested